# Nutritional Counseling Based on Mindful Eating for the Eating Behavior of People Living with Overweight and Obesity: A Randomized Clinical Trial

**DOI:** 10.3390/nu16244388

**Published:** 2024-12-20

**Authors:** Maria Fernanda Souza Moreira, Brenda Eduarda Fabris de Azevedo, Mileni Vanti Beretta, Fernanda Michielin Busnello

**Affiliations:** 1Graduate Program in Nutritional Sciences, Federal University of Health Sciences of Porto Alegre (UFCSPA),Porto Alegre 90050-170, RS, Brazil; mfermoreira@gmail.com (M.F.S.M.); mileni.nutri@gmail.com (M.V.B.); 2Undergraduate Program in Nutrition, Federal University of Health Sciences of Porto Alegre (UFCSPA), Porto Alegre 90050-170, RS, Brazil; brenda99fabris@gmail.com; 3Department of Nutrition, Federal University of Health Sciences of Porto Alegre (UFCSPA), Sarmento Leite Street, 245, Building, Historic Center, Porto Alegre 90050-170, RS, Brazil

**Keywords:** mindful eating, feeding patterns, weight management, behavioral intervention

## Abstract

Objectives: This study aimed to evaluate the effect of a counseling approach based on Mindful Eating on the eating behavior of individuals living with overweight and obesity. Methods: A 16-week randomized clinical trial was conducted to compare an online group intervention (IG) with individual online nutritional counseling (TAU). Volunteers (*n* = 61), aged 18 or older with BMI ≥ 25 kg/m^2^, were randomly assigned to the two groups and invited to participate in eight biweekly sessions. Eating behavior was assessed using the Three-Factor Eating Questionnaire (TFEQ-21), and anthropometric measures were collected through self-report. Comparison between the groups was performed using generalized linear models and Poisson models with robust variance estimators. A sensitivity analysis was conducted to evaluate the group-by-time interaction. The study was approved by the UFCSPA Ethics Committee. Results: At baseline, the eating behavior domain with the highest mean score was emotional eating (EE) in both groups. At the end of the study, all domains decreased, while an increase in cognitive restraint (CR) was observed in the TAU group, showing a difference between the groups (−23.1; 95%CI −37.7 to −8.5; *p* = 0.004). A group-by-time interaction difference was noted only in the CR domain, explained by the change in the TAU group for the CR comparison before and after treatment and by the difference between the IG and TAU groups at the end of the treatment period. Conclusions: A Mindful Eating approach may aid in managing dysfunctional cognitive restraint, a key component for sustainable excess weight treatment. The protocol can be found on ClinicalTrials.gov (NCT05845411).

## 1. Introduction

Obesity represents a significant risk factor for various chronic diseases [1]. When combined, these diseases reduce the quality of life, increase preventable early mortality, and burden healthcare systems [2]. Its multifactorial etiology involves genetic, physiological, cultural, environmental, socioeconomic, and political aspects [3] that extend beyond individual food choices and require coordinated actions to effectively promote prevention, management, and control [4].

Cognition and emotion guide eating behaviors, reflecting the interaction between psychological, physiological, and external environmental states in which we live [5]. Food-seeking behaviors influenced by psychological factors such as stress, anxiety, and depression have garnered interest, as they are associated with weight gain when they occur frequently over time [6].

Evidence suggests that emotional eating affects food choices [7] and the amount consumed, increasing the tendency toward uncontrolled eating [8]. This increased consumption of energy-dense foods is often associated with a preference for sweet and fatty foods [7]. Emotional eating is considered an unconscious attempt to regulate emotions that individuals are unable to manage in the moment [8]. However, it appears that this behavior does not affect all individuals equally; those with cognitive restraint—those who impose self-restrictions on food—seem to be more vulnerable to increased food intake in response to negative emotions [9,10]. In these cases, such emotions may impair cognitive control, leading to a loss of food self-regulation, also known as eating disinhibition [10].

Currently, some clinical guidelines for treating overweight continue to emphasize strict food restriction strategies. This narrow focus on specific eating patterns, often disconnected from individuals’ realities, can hinder their autonomous food choices and understanding of a health-promoting diet. Moreover, it hinders the establishment of sustainable changes in eating behavior [11]. Such approaches are often associated with weight regain [12], body dissatisfaction, and unhealthy eating behaviors [10]. A smaller number of guidelines adopt a more patient-centered approach, focusing on food as a whole rather than nutrients, and incorporating strategies that consider the influence of eating behavior on overweight [13,14,15].

Mindful Eating is a branch of mindfulness that aims to improve problematic eating behaviors and body image dissatisfaction [16,17] rather than focusing solely on weight loss. Through meditation and specific practice, it encourages increased awareness of the processes involved in food choices (e.g., recognizing patterns, automatic thoughts, or emotional cues that trigger eating) [16,18], and enhances the perception of bodily signals and sensations [19]. Initially, Mindful Eating was applied in the treatment of binge eating, leading to reduced binge eating severity and depression symptoms, and improved emotion regulation and mindfulness levels, compared to other psychotherapies or no intervention [20].

Although expanding the use of Mindful Eating as a strategy to manage problematic eating behaviors in individuals with excess weight seems promising, combining restrictive diets with the principle of guiding eating through hunger and satiety cues contradicts their core philosophy [10]. Furthermore, integrating mindfulness-based interventions with other strategies aimed at addressing disordered eating behaviors or weight management often obscures their independent effects [21]. This lack of clarity arises because lifestyle programs, while effective for short-term weight loss, typically rely on the temporary control of restrictive eating behaviors [22]. For individuals struggling with excess weight, adhering to a restrictive eating plan while simultaneously focusing on food awareness can be both challenging and confusing.

However, the use of Mindful Eating, among other mindfulness-based interventions adapted to the food context, as a complementary strategy for managing problematic eating behaviors in the treatment of overweight, has been tested in several studies [23,24,25]. These results are influenced by the different methods of intervention delivery and the type of treatment offered to the comparison group, promoting varied effects on different problematic eating behaviors [16,26]. The Mindful Eating program, as an adjunctive treatment for overweight in low-income women, compared to another mindfulness-based programs or no intervention, improved binge eating behavior [27]. Among overweight adults, it was effective in reducing emotional eating and external eating cues at the end of the program and at a 12-month follow-up [28]. A systematic review comparing Mindful Eating with or without diet with usual treatment, waitlist, and information-only programs found similarly satisfactory results for impulsive eating [29] and concerns about weight and shape, though findings were inconsistent for outcomes related to restrained eating [26]. When compared to waitlist conditions as a program for improving eating behavior in adults with overweight or with obesity, Mindful Eating associated with self-compassion promoted greater oral control and improvements in behavioral and cognitive characteristics typically observed in anorexia and bulimia nervosa [30].

Given the evidence of positive outcomes across certain problematic eating behaviors when used as a complementary treatment, it remains to be understood whether Mindful Eating, as a standalone treatment, is equally beneficial. We believe that this knowledge may be useful in organizing the different stages of treatment for overweight individuals suffering from problematic eating behaviors, especially those who have previously failed treatment. Therefore, we hypothesize that Mindful Eating as a single strategy can effectively improve problematic eating behavior in overweight individuals compared to conventional dietary treatment. Given the above, the objective of this study was to evaluate the effect of a nutritional intervention based on Mindful Eating, on changing the problematic eating behavior of individuals living with overweight and obesity.

## 2. Materials and Methods

### 2.1. Study Design and Participants

A parallel randomized clinical trial was conducted from July 2021 to July 2022 to compare the effectiveness of a Mindful Eating (ME)-based nutrition counseling group intervention with individual standard dietary treatment (TAU). Due to the COVID-19 pandemic and associated lockdown measures, the entire study was conducted in a virtual environment using a video conferencing platform hosted within the university’s domain. The virtual sessions were not recorded, and access was restricted to study participants and the researcher conducting the session, ensuring the confidentiality of both audio and video content.

The inclusion criteria were individuals of both sexes, over 18 years of age, living with overweight or obesity (Body Mass Index—BMI ≥ 25 kg/m^2^), who had access to the internet and were available to participate in eight sessions approximately every 15 days (with a maximum duration of four months) via video call. The exclusion criteria included reporting any cognitive, neurological, or psychiatric condition that would prevent participants from understanding or completing the activities proposed by the study, pregnancy or lactation, use of medication for obesity treatment, or previous bariatric surgery.

The sample size was estimated at 60 participants (30 per group) using the G*Power 3.1 software for an ANCOVA analysis. A significance level of 5% and a power of 80% were considered to detect a significant difference between groups, similar to the findings of Palmeira et al. [31], for the domains of eating behavior: emotional eating (f = 0.39) and uncontrolled eating (f = 0.45). We used the calculation obtained for the emotional eating domain due to the higher sample size, adding 10% to account for potential dropouts.

### 2.2. Ethical Aspects

This research was approved by the Research Ethics Committee of the Federal University of Health Sciences of Porto Alegre (CEP/UFCSPA) under number CAAE 46133421.1.0000.5345 after a complete review. The study protocol was registered at ClinicalTrials.gov (NCT05845411) and the Brazilian Clinical Trials Registry (RBR-2rkhnzt). All participants provided signed Informed Consent Forms (ICFs). The study was conducted in accordance with the ethical guidelines of the Declaration of Helsinki.

### 2.3. Study Procedure

The invitation to participate in the study was published on social media, with a link to complete the online screening form. Participants’ data, including the inclusion and exclusion criteria, were accessed through this form. The first 60 eligible volunteers were invited for an initial virtual interview with the principal investigator to obtain the consent to participate in the study, and baseline data were collected through an initial research questionnaire. Post-intervention data collection occurred in the last virtual session.

### 2.4. Measurements

The primary outcome assessed change in eating behavior across the following domains: uncontrolled eating (UE), cognitive restriction (CR), and emotional eating (EE). Secondary outcomes included anthropometric measurements of Body Mass Index (BMI).

### 2.5. Sociodemographics and Anthropometric Measurements

The demographic data collected included details on age and sex. Anthropometric weight and height measurements were self-reported, and BMI was subsequently calculated. To standardize the weight measurement, all volunteers received the following instructions before the two collection moments: measurement on the same scale and time (preferably in the morning), barefoot, with as little clothing and accessories as possible [32], placing the arms along the body and looking at a fixed point in front.

### 2.6. Eating Behavior Measurements

The abbreviated version of the 21-item Three-Factor Eating Questionnaire (TFEQ-R21), translated into Brazilian Portuguese and with previously evaluated psychometric properties [33], was used. This self-assessment scale is widely employed to estimate three aspects of eating behavior: cognitive restriction (CR) (six items that assess conscious cognitive control over eating to manage weight), uncontrolled eating (UE) (nine items that evaluate the tendency to lose control while eating, triggered by hunger or external stimuli), and emotional eating (EE) (six items that measure susceptibility to overeating under conditions of emotional stress or negative mood). The raw average of each behavioral dimension was converted into a score ranging from 0 to 100 to express the results. A higher score indicates a greater presence of that particular eating behavior domain in the individual’s eating habits [33].

### 2.7. Physical Activity Measurement

The assessment of the level of physical activity was performed using the International Physical Activity Questionnaire (IPAQ), in its short version, validated simultaneously in several countries, including Brazil [34]. The tool estimates the level of physical activity in a typical week through the self-reporting of the frequency and duration of walks and activities that require moderate and vigorous effort. The questionnaire consists of eight questions, and the final scores of the activity duration and frequency variables were obtained by adding the reported responses. For the final classification of the scores, we considered whether or not the individual reached the current recommendations for physical activity [35], and categorized them into sedentary, irregularly active A (reaches the recommended weekly time or frequency), irregularly active B (performs some activity, but does not reach any of the recommendations), active, or very active [34].

### 2.8. Interventions

Randomization in variable blocks of variables was performed on the website randomization.com (Boston, MA, United States [http://www.jerrydallal.com/random/randomize.htm accessed on 27 October 2021]) [36] using a 1:1 ratio. All researchers involved in participant allocation were blinded. Volunteers were randomly assigned to one of two groups: nutritional counseling based on Mindful Eating (ME) or standard individual dietary treatment (TAU). Both groups received eight virtual sessions, with approximately 15 days between sessions, over a period of up to four months.

The ME group was divided into subgroups of up to 15 people to facilitate interaction between participants, with sessions lasting up to two hours. Each session focused on a specific aspect of Mindful Eating, sometimes combined with content on eating behavior. The methodologies used included content exposition, large group discussions, and guidelines for informal Mindful Eating exercises to be incorporated into participants’ routines after the sessions. Examples of these practices are the application of the BASICS acronym (B: breathe and check your belly for signs of hunger before eating; A: assess your food; S: slow down; I: investigate your hunger during a meal, especially in the middle of the meal; C: chew food well; S: savor your food.) [37], practicing attention during culinary activities, and becoming aware of thought patterns during meals [38]. In addition to the sessions, the participants received visual materials and audio recordings of mindfulness meditation via cell phone message. All participants were encouraged to practice daily meditation.

The control group received traditional online nutritional monitoring and individual consultations lasting up to 1 h. After the initial appointment, an individualized eating plan was provided, designed to promote a caloric deficit of 500 to 750 kcal/day, with appropriate macronutrient distributions of 55 to 60% carbohydrates, 15 to 20% proteins, and 20 to 25% fats [14,39,40,41]. From the second appointment, in addition to assessing adherence and making the necessary adjustments to the treatment plan, food and nutrition education sessions were conducted based on the Food Guide for the Brazilian Population [42] (Appendix A), and virtual folders with nutritional guidelines were provided.

### 2.9. Data Analysis

Categorical variables were described as absolute and relative frequencies, while quantitative variables were described as mean and standard deviation (SD) or median and interquartile range (IQR) for skewed distributions. The distribution of each variable was assessed visually through histograms and based on its nature. Baseline comparisons between groups were performed using Student’s *t*-test for normally distributed continuous variables and the chi-square test for categorical variables.

Continuous outcomes were compared between the proposed treatments using generalized linear models, considering normal distribution, in a univariable manner at each time point and after intervention, and adjusted for baseline variables. Additionally, for each outcome measured before and after the intervention, a specific model was constructed to analyze the differences between the time points, enabling direct comparisons between the treatment groups. Model residuals were examined visually through histograms to ensure appropriate fit. We also use generalized linear mixed models to compare outcomes, incorporating the interaction between treatment groups and collection periods, followed by Tukey’s multiple comparisons test.

Adherence to the program and the impact of exclusions were compared between groups using generalized linear models with a Poisson distribution and robust variance estimator. There were no missing data for patients who completed the treatment. A significance level of 5% was used for all analyses. All analyses were conducted using R software, version 4.3.2 [43], an open-source statistical programming language widely used in clinical and epidemiological research for its flexibility, reproducibility, and extensive library of packages tailored to advanced statistical analyses.

## 3. Results

A total of 118 people applied to participate in the study, and 102 met the eligibility criteria and were invited to the initial interview. Of the 64 who attended the interview, 61 agreed to participate in the study and were randomized into two groups: 31 in the intervention group and 30 in the control group (Figure 1).

Table 1 presents baseline characteristics of the participants. In both groups, there were more women in an obese nutritional state and with more pronounced emotional eating behaviors.

Withdraw rates did not differ between groups (ME: 32.3% [*n* = 10]; TAU: 20% [*n* = 6]; risk ratio 1.61; 0.67 to 3.88; *p* = 0.286). Reasons for withdrawing from the study included returning to in-person work (ME: *n* = 5; TAU: *n* = 2) and being unable to attend the session or individual consultation time (ME: *n* = 3). Some participants were unresponsive to contact and missed their schedule session (ME: *n* = 2; TAU: *n* = 4). No difference was observed between those who completed the study and those who dropped out, across any variable.

### Effect of Mindful Eating

Table 2 shows the TFEQ scores for each behavior before and after treatment. Intragroup changes were observed after the intervention period in both groups.

In the ME group, UE decreased (−14.6; 95% CI −22.7 to −6.6; *p* = 0.001) with EE (−18.5; 95% CI −27.7 to −9.4; *p* < 0.001), while in the TAU group, CR significantly increased (17.1; 95% CI 7.4 to 26.9; *p* = 0.001).

Post-treatment TFEQ scores for each eating behavior domain, as well as the treatment effect adjusted for baseline, age, sex, BMI, and physical activity level variables, are also presented in Table 2. After treatment, a significant reduction in BMI was observed within the TAU group only, while the ME group did not show a statistically significant change. Regarding CR, the primary outcome, the ME group had a mean score 21.9 points lower (95% CI −32.5 to −11.4; *p* < 0.001) compared to the TAU group, while the TAU group showed a significant increase in CR score from the baseline. Additionally, the ME group had a mean BMI 4.2 kg/m² higher (95% CI 1.2 to 7.3; *p* = 0.009) compared to the TAU group. When adjusted for confounders such as age, sex, baseline BMI, and physical activity level, the post-treatment BMI difference was no longer significant (β = 0.80; 95% CI −0.06 to 1.6; *p* = 0.077). Similarly, no significant difference in BMI change from the baseline to the end of treatment was observed between groups (β = 0.66; 95% CI −0.17 to 1.4; *p* = 0.135). CR was the only eating behavior domain with a significant difference between groups after treatment. The sensitivity analysis revealed a significant group-by-time interaction effect on CR behavior (*p* < 0.001), which was explained by the differences observed in pre- and post-treatment scores in the TAU group (*p* = 0.003) and by the post-treatment comparison between the ME and TAU groups (*p* < 0.001).

## 4. Discussion

This study tested a collective nutritional intervention based on Mindful Eating as a standalone strategy for addressing problematic eating behaviors in individuals with overweight and obesity. Our hypothesis was confirmed only for the CR eating behavior, as evidenced by a significant difference in the final scores between the intervention and TAU groups. This difference was due to a significant increase in CR in the TAU group, which was treated with a hypocaloric eating plan combined with nutrition education sessions based on the Food Guide for the Brazilian Population [42] (Appendix A). Even with a more inclusive approach for the TAU group, our results highlight as a practical implication the need for systematically using strategies to manage rigid dietary restriction behavior in the treatment of overweight individuals. In contrast, the EE and UE domains did not show significant improvements beyond those achieved by a weight reduction program based on lifestyle changes, leading to the rejection of our hypothesis for these behaviors.

Similarly, the EATT study [30] demonstrated improvements in the behavioral and cognitive features typically observed in eating disorders (ED) (*p* = 0.046), but not in the dieting behavior subscale (group effect *p* = 0.071), after an 8-week Mindful Eating intervention without a dietary plan, in a Greek community sample of overweight individuals without ED. It is known that although CR is a predictor of weight loss in the short term [22], a stricter pattern of food restraint has been associated with disordered eating behaviors [44], reduced psychological well-being [45], greater adiposity [46], and worse maintenance of lost weight [47]. Additionally, most studies demonstrating successful weight reduction from restrictive eating strategies have limited follow-up periods [48] or show progressive weight regain over five years [49].

EE and UE were significantly reduced after the intervention. However, the effect of ME compared to TAU did not differ in changing these behaviors. In contrast, Morillo-Sarto et al., 2023 [28], demonstrated, through a community-based randomized clinical trial, a significant reduction in EE (β = −0.27; *p* = 0.006; d = 0.35), sustained at a 12-month follow-up (β = −0.53; *p* < 0.001; d = 0.69) for overweight individuals who received a combined ME and TAU intervention, compared to TAU alone. A similar result was reported in a systematic review of randomized controlled trials (RCTs) [26] that used mindfulness-based interventions to treat individuals exhibiting disordered eating behaviors, body image con-cerns, obesity or overweight. In addition to significant reductions in EE, participants in ME groups showed improvement in external eating, binge eating, and weight and shape concerns [26].

Lower EE in the intervention group can suggest less susceptibility to internal cues that lead to eating [50] and to palatable food-seeking behavior to regulate difficult emotional states [51], while a lower UE may indicate a better response to hunger and satiety signals and improved oral control [30]. Some poorly understood mechanisms involved in the consumption of hyper-palatable foods rich in carbohydrates and sodium make these foods targets of a hedonic or rewarding diet, facilitating excessive passive consumption and a high caloric density [52]. In this study, the ME intervention likely contributes to the development of the ability to accept negative emotions rather than suppress them through binge eating or food cravings. Previous findings have demonstrated a reduction in the association between negative mood and food cravings (β = −0.20; 95% CI −0.38 to −0.03; *p* = 0.021) in overweight women after a Mindful Eating intervention via a mobile device.

There was no difference in BMI change when comparing the results between groups at the end of the study. These findings are consistent with previous meta-analyses, which demonstrated a similar reduction in BMI [53] and waist circumference [54] from Mindfulness-based eating strategies compared to conventional diet programs or nonintervention controls, respectively. Mercado et al., 2021 [53], identified an insignificant difference of −0.24 kg/m² (95% CI −0.74 to 0.26; *p* = 0.34) between individuals with overweight or binge eating disorder (clinical and subclinical populations) after the intervention. However, additional results suggest that shorter interventions tend to have a significant effect, which diminishes as the intervention duration increases. It is important to note that this finding comes from studies that combined ME with diet or offered ME alone, leaving uncertainty about the real effect of ME. According to Fuentes Artiles et al., 2019 [54], in addition to the programs showing no difference in BMI change (−0.137 kg/m²; 95% CI −0.365 to 0.091; *p* = 0.240), no significant impact was observed on waist circumference reduction (−0.358 cm; 95% CI −0.916 to 0.200; *p* = 0.209). Although the systematic review by Carrière et al., 2018 [16], showed findings consistent with the others, it demonstrated that while controls gained some weight at follow-up, subjects in the ME group continued to lose weight.

Some limitations were identified in this study that deserve attention for future research. First, part of the intervention period coincided with the return to in-person activities after the COVID-19 pandemic in Brazil, which led to some volunteers being unable to continue participating in the study. Despite this, the attrition rate was similar to that of previous studies [16,55]. Second, given the non-representative sample of the target population, these results should be interpreted with caution and have limited generalizability. Third, anthropometric measurements were self-reported and subject to measurement bias. Additionally, the association between overweight individuals and the under-reporting of their current weight is well known. Fourth, to better understand gaps in eating behavior through Mindful Eating intervention studies, more components should be assessed (e.g., distractibility, awareness, acceptance, and mindfulness level) beyond the domains we measured with the TFEQ-R21. Finally, our group developed the intervention used, which did not constitute an official Mindful Eating program, limiting the comparison of our findings with those of other studies.

The Mindful Eating intervention proved effective in improving dietary restraint compared to conventional nutritional treatment, suggesting it may be a helpful approach for patients struggling with restrictive eating behaviors, which can lead to disordered eating patterns. However, the theoretical hypothesis that Mindful Eating may be sufficient for managing problematic eating behaviors in overweight individuals was not fully supported by this study. It appears that well-designed and inclusive lifestyle programs may have similarly positive effects on both binge eating and emotional eating behaviors. Therefore, it is clear that different problematic eating behaviors are likely to respond to distinct strategies. Additionally, its limited effectiveness in reducing BMI suggests that more structured dietary guidance may be needed for weight management. Future studies could explore the impact of a more intensive or continued Mindful Eating intervention (with maintenance sessions) in groups with greater gender diversity and clinical conditions, to better understand the scope and limitations of this intervention. Larger samples that allow for stratification of individuals into subgroups based on BMI for overweight and obesity are also important.

## 5. Conclusions

After four months of a single Mindful Eating intervention, people living with excess weight demonstrated significantly less cognitive restraint than those enrolled in conventional diet programs. While Mindful Eating was able to stabilize this behavior, conventional programs exacerbated cognitive restraint in dieting. However, emotional eating and uncontrolled eating appear to be equally responsive to well-designed conventional eating programs that provide food and nutrition education. Our findings reinforce that a strategy based on Mindful Eating can be an important ally in managing problematic eating behaviors, a key component of overweight treatment. However, additional studies are needed to determine whether it is best used as a complementary or alternative strategy and to assess its effects on long-term problematic eating behavior of individuals with overweight. 

## Figures and Tables

**Figure 1 nutrients-16-04388-f001:**
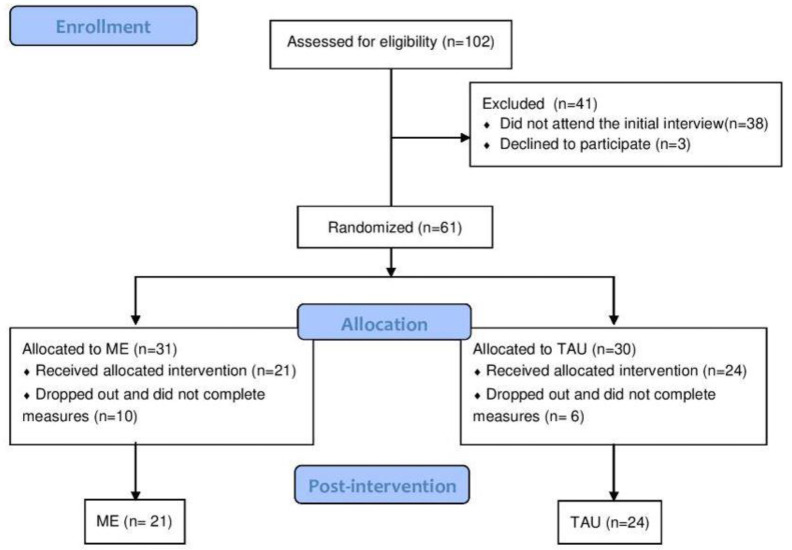
Flowchart of study participants. ME—intervention based on Mindful Eating; TAU—treatment as usual; IPAQ—International Physical Activity Questionnaire.

**Table 1 nutrients-16-04388-t001:** Baseline characteristics by group (*n* = 61).

Variable	ME	TAU	*p*-Value
	*n*, %	*n*, %	
	Mean (SD)	Mean (SD)	
	(*n* = 31)	(*n* = 30)	
Age (years)	35.8 (10.8)	37.43 (11.04)	0.571
Sex			
Male	2 (6.5%)	5 (16.7%)	0.255
Female	29 (93.5%)	25 (83.3%)	
BMI (kg/m²)	34.6 (5.9)	31.9 (4.5)	0.05
TFEQ-R21 domain			
Uncontrolled Eating	55.4 (19.2)	52.8 (16)	0.569
Cognitive Restraint	51.8 (16.2)	52.6 (18.4)	0.858
Emotional Eating	67.6 (22.6)	60.2 (24.9)	0.23
IPAQ			
Very Active	7 (22.6%)	6 (20%)	0.601
Active	17 (54.8%)	13 (43.3%)	
Irregularly Active—A	2 (6.5%)	5 (16.7%)	
Irregularly Active—B	2 (6.5%)	4 (13.3%)	
Sedentary	3 (9.7%)	2 (6.7%)	

ME = intervention based on Mindful Eating; TAU = treatment as usual; BMI = Body Mass Index; TFEQ-R21 = Three-Factor Eating Questionnaire; IPAQ = International Physical Activity Questionnaire; significance was set at *p* < 0.05.

**Table 2 nutrients-16-04388-t002:** Evaluation of the effect of the intervention at the end of the study using generalized linear models.

	ME (*n* = 21)	TAU (*n* = 24)	Adjusted Model ***	*p*-Value
	Mean (SD)/Difference Mean (95%CI)	Mean (SD)/Difference Mean (95%CI)	β (95%CI)	
TFEQ-R21 domain				
Uncontrolled Eating				
Baseline score	54.9 (20.4)	53.4 (17)	2.6 (−9.2; 14.5)	0.662
Post-treatment score	40.2 (18.2)	32.1 (14.3)	7.4 (−1.9; 16.7)	0.126
Score difference from baseline	−14.6 (−22.7; −6.6) *	−21.3 (−29.9; −12.7) **	5.9 (−5.3; 17.2)	0.308
Cognitive Restraint				
Baseline score	49.7 (17.4)	50.9 (19.4)	1.4 (−10.8; 13.6)	0.803
Post-treatment score	46.6 (18.5)	68.1 (13.1)	−21.9 (−32.5; −11.4)	<0.001
Score difference from baseline	−3.1 (−12.7; 6.3)	17.2 (7.4; 26.9) *	−23.1(−37.7; −8.5)	0.004
Emotional Eating				
Baseline score	67.2 (23.7)	58.8 (26.8)	6.2 (−10.9; 23.3)	0.483
Post-treatment score	48.7 (23)	42.6 (23.6)	3.3 (−7.9; 14.6)	0.567
Score difference from baseline	−18.5 (−27.7; −9.4) **	−16.2 (−25; −7.4) **	0.93 (−11.9; 13.8)	0.888
BMI (kg/m²)				
Baseline	35.1 (5.8)	31.5 (4.4)	3.9 (0.87; 7.0)	0.016 *
Post-treatment	34.5 (5.6)	30.3 (4.7)	0.8 (−0.06; 1.6)	0.077
Difference from baseline	−0.6 (−1.3; 0.1)	−1.3 (−1.7; −0.8) *	0.6 (−0.18; 1.4)	0.135

ME = intervention based on Mindful Eating; TAU = treatment as usual; BMI = Body Mass Index; TFEQ-R21 = Three-Factor Eating Questionnaire. Significance is set at * *p* < 0.05 and ** *p* < 0.001. *** Model adjusted for age, gender, BMI, and initial physical activity and initial domain.

## Data Availability

Data available in a publicly accessible repository (https://osf.io/er72q/?view_only=d39f55cb1d954887ba6c595db26e4cdd accessed on 15 November 2024 or DOI 10.17605/OSF.IO/ER72Q).

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
