# Peer review of "Nutritional Counseling Based on Mindful Eating for the Eating Behavior of People Living with Overweight and Obesity: A Randomized Clinical Trial"

_nutrients, 2024, doi:10.3390/nu16244388_

Round 1

Reviewer 1 Report

Comments and Suggestions for Authors

This paper is well-written and positioned. However, there are some corrections as follows.

1. In the introduction, the key contribution of this study compared to existing research is ambiguous. Please describe the selling point of this study from a theoretical perspective.

2. Randomized clinical trials were conducted during the pandemic. We need a rationale for whether virtual environments are appropriate.

3. The two groups are presented in Tables 1 and 2. Are the differences between the groups statistically significant?

4. The authors should clearly state the theoretical and practical implications based on their findings.

Reviewer 2 Report

Comments and Suggestions for Authors

The article entitled „Nutritional counseling based on Mindful Eating for the eating behavior of people living with overweight and obesity: a randomized clinical trial” is an important work indicating the need for both a dietitian and a psychologist in the treatment of overweight and obesity.

The text of the article was carefully prepared in compliance with good practices in writing a research paper.

Only minor comments to consider:

- selecting one of the statements "Mindful Eating" or "Mindfulness" in keywords,

- making the record of analytical methods legible in relation to the presented data - the record is too general and uses language that is unclear to the recipient,

- explaining the R software,

- considering leaving only one form of results in Table 2 (model adjusted) without graphs 2, 3, 4; supplementing the table with initial data,

- starting the discussion by fulfilling or rejecting the hypothesis.

Round 2

Reviewer 1 Report

Comments and Suggestions for Authors

The correction is good. Thus, I would like to accept the revision work.